# Integrative Analysis Reveals the Expression Pattern of SOX9 in Satellite Glial Cells after Sciatic Nerve Injury

**DOI:** 10.3390/brainsci13020281

**Published:** 2023-02-07

**Authors:** Kuangpin Liu, Wei Ma, Jinwei Yang, Wei Liu, Sijia Zhang, Kewei Zhu, Jie Liu, Xianglin Xiang, Guodong Wang, Hongjie Wu, Jianhui Guo, Liyan Li

**Affiliations:** 1Institute of Neuroscience, Faculty of Basic Medical Science, Kunming Medical University, Kunming 650500, China; 2Second Department of General Surgery, First People’s Hospital of Yunnan Province, Kunming 650032, China

**Keywords:** integration analysis, sciatic nerve injury, satellite glial cell, Sox9 expression pattern

## Abstract

Background: Several complex cellular and gene regulatory processes are involved in peripheral nerve repair. This study uses bioinformatics to analyze the differentially expressed genes (DEGs) in the satellite glial cells of mice following sciatic nerve injury. Methods: R software screens differentially expressed genes, and the WebGestalt functional enrichment analysis tool conducts Gene Ontology (GO) enrichment and Kyoto Encyclopedia of Genes and Genomics (KEGG) pathway analysis. The Search Tool for the Retrieval of Interacting Genes/Proteins constructs protein interaction networks, and the cytoHubba plug-in in the Cytoscape software predicts core genes. Subsequently, the sciatic nerve injury model of mice was established and the dorsal root ganglion satellite glial cells were isolated and cultured. Satellite glial cells-related markers were verified by immunofluorescence staining. Real-time polymerase chain reaction assay and Western blotting assay were used to detect the mRNA and protein expression of Sox9 in satellite glial cells. Results: A total of 991 DEGs were screened, of which 383 were upregulated, and 508 were downregulated. The GO analysis revealed the processes of biosynthesis, negative regulation of cell development, PDZ domain binding, and other biological processes were enriched in DEGs. According to the KEGG pathway analysis, DEGs are primarily involved in steroid biosynthesis, hedgehog signaling pathway, terpenoid backbone biosynthesis, American lateral skeleton, and melanoma pathways. According to various cytoHubba algorithms, the common core genes in the protein–protein interaction network are Atf3, Mmp2, and Sox9. Among these, Sox9 was reported to be involved in the central nervous system and the generation and development of astrocytes and could mediate the transformation between neurogenic and glial cells. The experimental results showed that satellite glial cell marker GS were co-labeled with Sox9; stem cell characteristic markers Nestin and p75NTR were labeled satellite glial cells. The mRNA and protein expression of Sox9 in satellite glial cells were increased after sciatic nerve injury. Conclusions: In this study, bioinformatics was used to analyze the DEGs of satellite glial cells after sciatic nerve injury, and transcription factors related to satellite glial cells were screened, among which Sox9 may be associated with the fate of satellite glial cells.

## 1. Introduction

A common clinical condition is peripheral nerve injury (PNI). PNI can result in motor and sensory dysfunctions, which burden families and society. The research has always aimed to promote PNI regeneration and repair [1]. Relevant research suggests that compensation or regeneration may be used to restore the damage. For example, the collateral branches of healthy neurons can promote nerve regeneration and functional recovery when the axon regeneration of injured neurons is blocked. Basic research has established a local sciatic nerve injury (SNI) model that further clarifies the compensatory growth mechanism and regeneration process [2].

There are two difficulties in treating nerve damage: How can neurons be regenerated to replace those lost to injury or neurodegeneration? How can secondary tissue damage (glial scar) that results from a long-term accumulation of glial cells at the injured site be prevented or minimized [3]? The ability of glial cells to regenerate differs significantly between the peripheral nervous system (PNS) and central nervous system (CNS). Virchow initially proposed the term “glial cell” in 1850. He believed that the term “glial cell” referred to a type of cell composed of neurons embedded in the connective tissue layer. Astrocytes, oligodendrocytes, and microglia are glial cells in the CNS, whereas Schwann cells (SCs) and satellite glia are found in the PNS [4]. The glial cells in the CNS offer the growth factors and structural support required for the regeneration of neurons by removing debris from the injured site and responding quickly to the nerve injury [5]. It can prevent the formation of a reactive glial scar, stimulate axon regeneration, and promote the recovery of motor, sensory, respiratory, and autonomic functioning in rodents after a spinal cord injury. The research demonstrates that astrocytes are helpful in spinal repair. The use of astrocyte transplantation in the treatment of spinal cord injury has grown in the past three decades [6]. A promising therapy for nerve regeneration and repair has recently emerged and is called in vivo glial neuron transformation technology. This is achieved by the ectopic production of neurotrophic factors in glial cells and converting endogenous glial cells into neurons [7]. The direct reprogramming of endogenous glial cells offers considerable potential for functioning neurons to repair the nerve injury, in contrast to the risk of immunological rejection and tumorigenesis when exogenous cells are transplanted to the location of the nerve injury site [8]. Astrocytes are promising source cells to replace neurons lost due to diseases since they have a common lineage and can differentiate and proliferate under pathological conditions. Astrocytes can be reprogrammed to become neurons by regulating transcription factors (TFs) [9]. For example, NeuroD1 can effectively reprogram gray matter astrocytes into functional neurons [7].

The PNS contains myelinated glial cells called SCs. Their plasticity is crucial to peripheral nerve regeneration following trauma and peripheral neuropathy. When a nerve is injured, SCs are quickly activated by the signal induced by the injury and begin the repair process in response. SCs perform dynamic cell reprogramming to promote nerve regeneration and functional recovery during the repair process. They actively promote neuron survival, damaged axon disintegration, myelin sheath clearance, and axon regeneration by expressing several new genes, which regulate and drive the regeneration process [10]. Treating nerve damage with SCs transplantation is reliable, effective, and promising. However, a single SCs transplant is insufficient to promote the complete recovery of neural function. More approaches are required to support SCs transplantation as a nerve injury treatment [11]. The TF c-Jun, which is quickly upregulated following SC injury, controls the reprogramming and repair process of SCs. The injury will result in dysfunction, neuronal death, and failure of functional recovery without c-Jun. Although c-Jun is unnecessary for developing SCs, it is crucial for reprogramming SCs to repair the damage [12]. Satellite glial cells (SGCs) are another type of glial cell in the PNS. In sensory ganglia, SGCs wrap sensory neurons and regulate their microenvironment and signal transmission [4]. SGCs are similar to astrocytes. They act as a buffer for the extracellular environment by expressing glial fibrillary acidic protein (GFAP) and potassium and calcium channels [13]. Additionally, both cell types serve as boundary cells in the nervous system, which can conduct intercellular signal transduction using chemical messengers and calcium waves [14]. Increasing intercellular coupling via gap connections, decreasing the expression of the inward rectifier potassium channel 4.1, and increasing the expression of GFAP and the p75 neurotrophin receptor (p75NTR), PNI-induced SGCs to appear reactive [13]. According to Matthias et al. [15], SGCs can be partially reprogrammed, which may be related to their traits and the plasticity of their precursors. The difficulty of conducting in vitro and in vivo experiments may be why little is known about the biology of SGCs compared to other glial cells in the CNS. Currently, neuropathic pain is the main topic of most studies on satellite glial cells, especially those involving chronic pain and inflammatory diseases. SGCs involved in neurogenesis, regeneration, and damage repair are rarely reported.

This study integrated and analyzed the gene expression profile data set in the Gene Expression Omnibus (GEO) database to identify differentially expressed genes (DEGs) in SGCs following SNI. We suggest using bioinformatics to mine core genes related to PNI and repair and focus on TFs in core genes. Because transcription factor-mediated cell reprogramming aids in understanding how glial cells mature into functional neurons and promote the recovery of neural function, it is an effective way for cell transformation during nerve regeneration [4]. The development of an SNI model to produce SGCs for verification offers a new clue for the peripheral glial cell replacement therapeutic strategy.

## 2. Materials and Methods

### 2.1. Data Extraction

Use the following search terms to find the results: “Satellite glial cells, SNI” (keywords), and “Mus musculus” (biology). The following were the inclusion criteria: (1) satellite glial cell samples of mice were diagnosed; (2) there were >3 samples in each group; (3) single-cell expression profile; (4) for satellite glial cells, there is only a single cause of SNI and no other treatment factors. Each group had to have >3 samples, and 114 gene set enrichment series were retrieved from the GEO database of the National Center for Biotechnology Information (https://www.ncbi.nlm.nih.gov/geo/; accessed on 16 June 2022). The original gene expression profile met these requirements for GSE120284, which was integrated and analyzed. This profile was sequenced using the gpl21103 Illumina HiSeq 4000 sequencing platform. Finally, eight samples were analyzed, including samples from a sham operation group (GSM 3978516, GSM 3978518, GSM 3978520, and GSM 3978522) and an injury group (GSM 3978501, GSM 3978503, GSM 3978505, and GSM 3978507) [16].

### 2.2. DEGs Analysis

Preprocessing of the original GSE120284 dataset included normalization and log2 conversion. The thermal heat map was then drawn based on the amount of DEGs, and a volcano plot was created based on the log2 fold change and *p* value (|log2FC| > 1, *p* < 0.05). R software was used to draw the figures.

### 2.3. Functional Enrichment Analysis of DEGs

Analysis of DEGs using the Gene Ontology (GO) and Kyoto Encyclopedia of Genes and Genomes (KEGG) was performed using the WebGestalt (http://www.webgestalt.org/, accessed on 16 June 2022) database. Genes related to molecular function, biological process, and cell composition were found using GO analysis. A signaling pathway-based visualization of the enrichment analysis is displayed.

### 2.4. Protein–Protein Interactions (PPIs) Analysis of DEGs

The PPIs of DEGs were analyzed, the gene data were integrated, and upregulated and downregulated genes were mapped via a PPI network diagram using the Search Tool for the Retrieval of Interacting Genes/Proteins (STRING) database (https://string-db.org/, accessed on 16 June 2022). Core genes were screened using Cytoscape (Ver 3.6, Beijing, China).

### 2.5. SNI Model in Mice

Five-day-old female C57/BL6 mice were purchased from the Department of Experimental Animals, Kunming Medical University, license no. SCX (Dian) K2020–0004. All experimental protocols were approved by the Animal Experiment Ethics Committee of Kunming Medical University. A 0.2 cm skin incision was made along the unilateral sciatic nerve trunk before injecting sodium pentobarbital 30 mg/kg (WuXi AppTec, Shanghai, China) intraperitoneally into the model animals. The muscle layer was bluntly separated with hemostatic forceps to expose the sciatic nerve trunk. The incision was then cut with surgical scissors. [17]. A total of 36 mice were randomly assigned to four groups (*n* = 9 per group): (1) Sham 3d group, (2) SNI 3d group, (3) Sham 7d group, and (4) SNI 7d group.

### 2.6. Cell Culture

C57BL/6 mice were disinfected with 75% ethanol for 10 min and sacrificed in cervical position; the mice in the sham operation group and the operation group were decapitated. The skin on the back and ribs of the mice were removed with scissors, along with the muscles surrounding the spine, exposing the spine. The blood vessels and spinal cord were cleaned, and the dorsal root ganglion (DRG) was clamped using ophthalmic micro tweezers. DRG were located at the L4 segment on the opposite side of the injured side. The nerve fibers on the DRG were removed and the capsule on the surface of the DRG was stripped. The capsule on the surface of the DRG was stripped, and the nerve fibers were removed. About 10 DRG were inserted in each well of a six-well plate containing DRG-SGCs culture medium and cultured in a 37 °C, 5% carbon dioxide incubator (Thermo Fisher Scientific, Waltham, MA, USA) [18].

### 2.7. Immunofluorescence Staining

The cell cultures were placed in six-well plates, washed with phosphate-buffered saline (HyClone, Logan, Utah, USA), mixed with 0.3% Triton X-100 (Sigma Aldrich, St. Louis, MO, USA), and then incubated at room temperature for 1 h. The primary antibodies used were glutamine synthetase (GS) (1:1000, Abcam, Cambridge, UK), GFAP (1:1000, Abcam, Cambridge, UK), p75NTR (1:1000, Abcam, Cambridge, UK), Nestin (1:1000, Abcam, Cambridge, UK), and SRY-box9 (Sox9) (1:1000, Abcam, Cambridge, UK). A combination containing the primary antibody and 5% goat serum albumin (1:1000, Santa Cruz, Dallas, Texas, USA) was combined and incubated at 4 °C overnight. The following day, the cell cultures were incubated at room temperature for 30 min, washed with phosphate-buffered saline with Tween 20 (PBST), and then incubated with a secondary antibody immunoglobulin G (1:1000, Abcam, Cambridge, UK). The second antibody was diluted with 5% goat serum albumin (Gibco, Grand Island, NY, USA) at room temperature for 2 h. First, cells were washed with PBST and stained with nuclear dye 4’, 6-diamino-2-phenylindole (DAPI) (1:1000, Sigma Aldrich, St. Louis, MO, USA), which was diluted with 2% goat serum albumin. Finally, the cells were washed with PBST. Images were captured using fluorescence microscopy (DS-Vi1 and Az100, Nikon, Tokyo, Japan). Image J (Ver 1.8.0, Bethesda, MD, USA) was used to count the number of positive immunofluorescent cells.

### 2.8. Real-Time (RT) Polymerase Chain Reaction Assay

SGCs were treated with the TRIzol solution (Invitrogen, Carlsbad, CA, USA) to extract their total RNA. Complementary (cDNA) was reverse-transcribed using SuperScript III (Takara, Osaka, Japan). GADPH was used as a negative control, and the SYBR quantitative qPCR kit (Takara, Osaka, Japan) was used to measure the relative expression of mRNA. The ABI Prism 7500 Rapid Sequence detection system (Applied Biosystems, Carlsbad, CA, USA) was used to perform qRT-PCR. The relative expression levels of mRNA were calculated and quantified using the 2^−ΔΔCt^ method. The primer sequence for Sox9 was as follows: F 5′-GTGCAAGCTGGCAAAGTTGA-3′, R 5′-TGCTCAGTTCACCGATGTCC-3′.

### 2.9. Western Blotting Assay

Total protein was extracted from SGCs, and each tube contained 200 μL of a detergent lysate (containing 2 μL of phenylmethylsulfonyl fluoride and 2 μL of phosphatase inhibitor). A microplate analyzer (DG-3022A, Tecan, Männedorf, Switzerland) was used to determine the protein concentration after diluting the samples. The extracted protein supernatant and 5× protein loading buffer (4:1, Solarbio, Beijing, China) were placed in boiling water for a 10-min denaturation process. After electrophoretic gel preparation, the primary antibodies were β-actin (1:500, Bioss, Beijing, China) and Sox9 (1:1000, Abcam, Cambridge, UK). We scanned the film for recovery value analysis (Bio-Rad, Hercules, CA, USA). Image J (Ver 1.8.0, Bethesda, MD, USA) was used to count gray value.

### 2.10. Statistical Analysis

Prism software (Ver 7.0, GraphPad Software, San Diego, CA, USA) was used for data analysis. All data are expressed as mean ± standard deviation (S.D.). Analysis of variance (ANOVA) was used, followed by Bonferroni post-hoc test between groups. *p* < 0.05 was considered statistically significant.

## 3. Results

### 3.1. Identification of DEGs in GSE120284

A total of 991 DEGs were identified by R software. Among them, 383 genes were upregulated, and 508 genes were downregulated (Figure 1a). A Pearson correlation analysis was used to create the clustering heat map. The data followed a normal distribution after undergoing logarithmic transformation, and hierarchical clustering was carried out. The clustering distance was averaged by the correlation coefficient matrix calculated by Pearson (Figure 1b).

### 3.2. GO and KEGG Pathway Enrichment Analysis of DEGs

The GO analysis revealed that DEGs are primarily enriched in biological processes related to sterol, cholesterol, secondary alcohol, and other biological processes. DEGs are mainly associated with negative regulation of cell development, mitochondrial ATP synthesis-coupled proton transport and basal plasma membrane, and other related cell components. DEGs are mainly related to PDZ domain binding, cell adhesion molecule binding, ubiquitin-like protein conjugating enzyme binding, and other molecular functions (Figure 2a,b).Additionally, the KEGG pathway analysis revealed that DEGs are primarily involved in the pathways for steroid biosynthesis, hedgehog signaling, terpenoid backbone biosynthesis, amyotrophic lateral sclerosis, and melanoma (Table 1).

### 3.3. PPI Network Construction and Core Genes Analysis

The STRING online tool was used to conduct a PPI network analysis of DEGs, which resulted in the identification of interactions between 849 nodes (proteins related to DEGs and their related proteins) and 3164 edges (differences between proteins related to DEGs and their related proteins) (Figure 3). Cytoscape software was used to screen the top 10 core genes; common core genes in different algorithms are Atf3, Mmp2, and Sox9, which are upregulated after SNI. Interestingly, the three core genes are all TFs (Table 2).

### 3.4. SNI Model and SGCs Culture

We located the sciatic nerve accurately (Figure 4) and built the mold. DRG cells were isolated and cultured from the injured contralateral side in a particular medium. Then, immunofluorescence was used to identify and culture the SGCs at 3d. The number of positive cells expressing SGC-specific markers GFAP and GS in the injury group was significantly higher than in the Sham group (Figure 5). The co-labeling results of GS and Sox9 are also shown, and they were significantly expressed in the injured group compared with the Sham group (Figure 6). Simultaneously, stem cell characteristic markers Nestin and p75NTR were labeled SGCs, and they were significantly expressed in the injured group compared with the Sham group (Figure 7). These findings suggest that SGCs can be labeled with Sox9, which may be a potential marker. In addition, stem cell markers Nestin and p75NTR can also label SGCs, suggesting that SGCs may have stem-like properties. 

### 3.5. Sox9 mRNA and Protein Expression

The results showed that mRNA expression of Sox9 in the SGCs injured group was upregulated compared to that in the Sham group at 3d and 7d (Figure 8). The protein expression of Sox9 in the SGCs injured group was upregulated compared to that in the Sham group at 7d (Figure 9). Collectively, these results suggest that mRNA and protein expression of Sox9 in SGCs were increased after SNI.

## 4. Discussion

SNI is the most prevalent PNS, characterized by motor and sensory fiber damage [19]. After SNI, cell structure and function change the spatial distribution of motor neurons and glial cells [20]. A study showed that glial cells in DRG play an important role in the model of neuropathic pain and chronic constriction injury of the sciatic nerve, but the specific role is still unclear. The regeneration process following SNI is related to biological activity in DRG. The regeneration process involves several gene expression changes, among which TFs play a crucial role [21]. The ability of TFs to induce cell differentiation, dedifferentiation, and transdifferentiation has been confirmed by many studies [22].

In this study, 991 DEGs, including 383 upregulated genes and 508 downregulated genes, were discovered by integrating and analyzing the information on the gene expression profile of SGCs following SNI. DEGs mostly concentrate on biosynthesis in biological processes. DEGs are mainly related to the negative regulation of cell development, mitochondrial ATP synthesis coupled with proton transport, and matrix membrane. Regarding molecular function, DEGs are associated with PDZ domain binding, cell adhesion molecule binding, a ubiquitin-like protein, coupling enzyme binding, etc. The KEGG analysis showed that DEGs were predominantly abundant in the biosynthetic pathway, inflammatory disease pathway, neurodegenerative disease pathway, and tumor signal pathway. 

The core genes Atf3, Mmp2, and Sox9, which are all upregulated and appear simultaneously according to the algorithm in the Cytohubba plug-in, are selected by Cytoscape after it analyzes the core genes in the PPI network.

Interestingly, the three core genes are TFs. TFs act as regulatory factors and select genes, determining cell types, development patterns, and specific pathways (such as immune response). According to the analysis of relevant literature, Atf3 does not exist in the DRG neurons of normal adult rodents. Still, it is exhibited in the sensory neurons of DRG after PNI and regulates axonal regeneration [23]. Lin JH et al. [24] concluded that the expression of Atf3 in neurons of neurofilament heavy chain-passive DRG in the acute phase was a potential biological hallmark of chronic pain in the lumbar radiculopathy rat model. Matrix metalloproteinases (Mmps) are inflammatory response proteins that regulate extracellular matrix remodeling, cell interaction, and signal transduction [25]. Mmp2 is a gelatinase with several functions at the neurovascular interface [26]. Mmp2 can be used as a molecular target of neuralgia in the DRG [27]. Sox9 is a member of the Sox TFs family, closely associated with stem cell biology, cell reprogramming, lineage, and differentiation [28]. In the CNS glial region, Sox9 accumulates during development. According to some research, Sox9 may regulate neurogenesis and mediate the switch from neurogenesis to gliogenesis [29,30]. Qiu B’s team [31] showed that functional astrocytes might be produced directly from fibroblasts by overexpressing the three TFs, NFIA, NFIB, and SOX9. These induced astrocytes to express GFAP and S100β, and other astrocyte markers. However, there are not many reports on Sox9 in PNS, which also aroused our interest.

Li’s team [32] found that three days after PNI, GFAP-positive cells started to surround DRG neurons, and these cells proliferated in vivo due to nerve injury. The progenitor cell markers p75NTR and Nestin are expressed simultaneously by these types of proliferating cells, which may represent SGCs. This is connected to the fact that, after PNI, the number of sensory neurons in DRG initially decreased but then returned to normal levels after a few months; however, the specific mechanism is yet unknown. Based on this, our team developed a primary culture method that produced very pure SGCs from rat DRG without digestion. The cells began to display the markers for neural crest progenitor cells, p75NTR, GFAP, and GS after three days of culture [18]. After SNI in mice, SGCs were isolated from the DRG using the culture method. Immunofluorescence verified that they expressed the glial cell markers GFAP and GS and the progenitor cell markers Nestin and p75NTR. 

In comparison to before the injury, there were more tagged positive cells after the nerve injury. SGCs, Sox9, and GS can be co-labeled; after nerve injury, there were more co-labeled positive cells than before.

Sox9 is a CNS astrocyte marker [33] associated with astrogenesis and whose expression in the brain affects both the development and survival of neuronal precursors and neurons [34]. Sox9-positive cells proliferate in the spinal cord parenchyma after spinal cord injury and participate in the development of a glial scar [35]. In this study, we found that Sox9-positive cells in SGCs proliferated after PNI, and Sox9 mRNA and protein expression increased by RT-PCR and Western blotting, which was consistent with the trend of Sox9 expression in integrated analysis. Sox9 was expressed in the PNS SGCs like the CNS astrocytes, and its expression level increased in response to nerve injury. SGCs can be labeled using Nestin and p75NTR. After PNI, the number of positive cells increases, consistent with previous studies suggesting that SGCs may be peripheral glial cells with stem cell characteristics [18]. This could be the difference between SGCs and astrocytes, or it could be the determinant of the difference in the ability of the CNS and PNS to regenerate differently.

Currently, the ideal treatment for SNI is autologous nerve transplantation. However, this approach compromises healthy nerves, necessitates a highly intensive surgery, and leaves room for other advanced transplantation options [36]. Using stem cells from neural stem cells, bone marrow, adipose tissue, and embryonic stem cells has emerged as a possible therapeutic approach for PNI. In animal models, inserting these cells into the severed sciatic nerve can trigger nerve regeneration and myelin synthesis. In the next few years, this treatment approach might become a conventional technology; however, further research is required [37]. Microarray technology is used in the immune response, acute inflammation, apoptosis, cell adhesion, ion transport, and extracellular matrix to screen DEGs of SNI. Interleukin-6, interleukin-1, integrin, c-sarcoma, cardiovascular antigen-related cell adhesion molecules, chemokine ligand, matrix metalloproteinase, etc., are important factors [38]. Studies suggest that regulating the inflammatory response is still effective in promoting peripheral nerve regeneration [16,38]. TFs might offer new clues for the study of SGCs and peripheral nerve regeneration.

## 5. Conclusions

In conclusion, this study used bioinformatics to analyze the gene expression profile in SGCs following SNI and to screen the TF Sox9 that may determine the fate of SGCs. Further investigation is required to understand how Sox9 regulates SGCs’ fate and mediates the transformation between neurogenic and glial origins in PNS.

## Figures and Tables

**Figure 1 brainsci-13-00281-f001:**
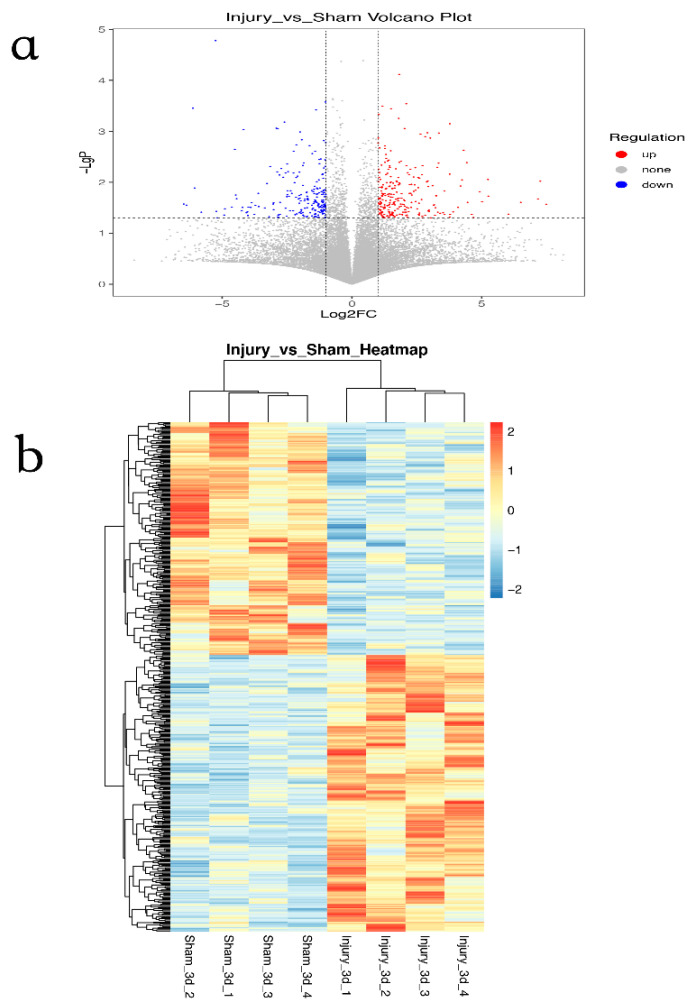
Identification of differentially expressed genes (DEGs) in GSE120284. (**a**) A volcano plot (Red: upregulated DEGs, Blue: downregulated DEGs); (**b**) Pearson correlation clustering heat map.

**Figure 2 brainsci-13-00281-f002:**
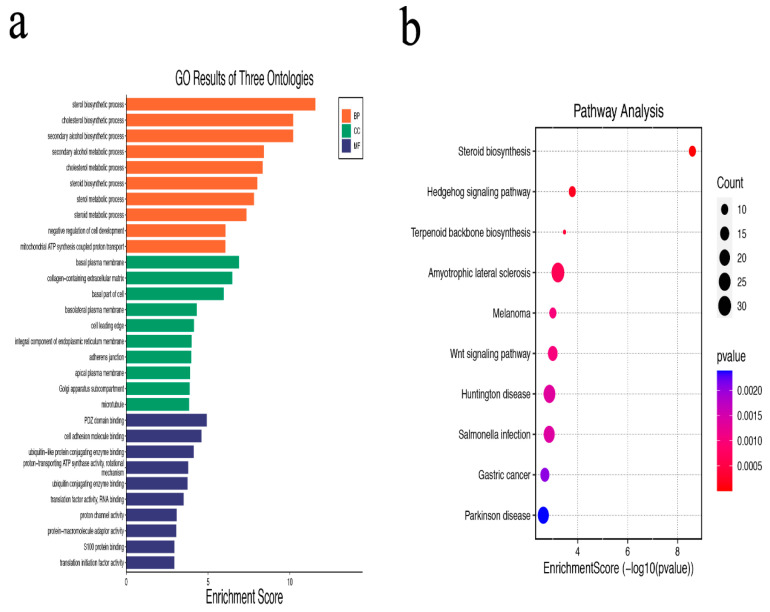
Gene ontology and Kyoto Encyclopedia of Genes and Genomics pathway enrichment analysis of differentially expressed genes in GSE120284 (Orange Bar: Biological Process; Green Bar: Cellular Component; Purple Bar: Molecular Function, the ordinate indicates the number of related genes). (**a**) Gene ontology analysis; (**b**) Kyoto Encyclopedia of Genes and Genomics pathway enrichment analysis.

**Figure 3 brainsci-13-00281-f003:**
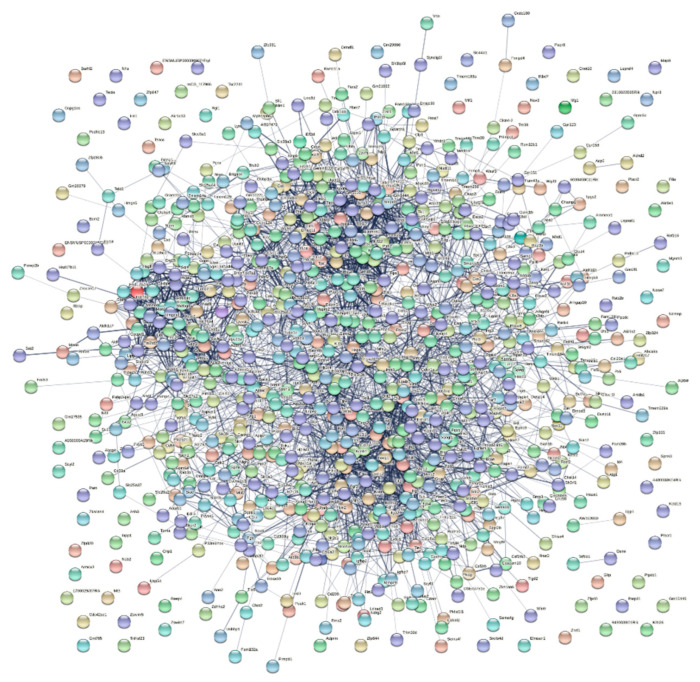
PPI Network Construction.

**Figure 4 brainsci-13-00281-f004:**
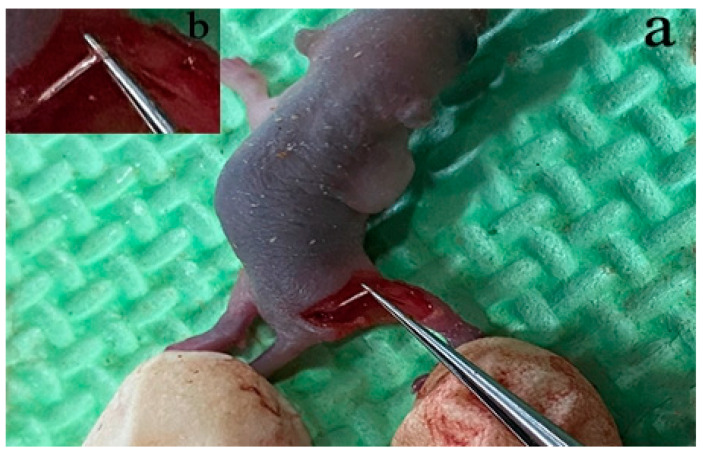
Locate the sciatic nerve of C57/BL6 (**a**) full image; (**b**) magnified image of the sciatic nerve.

**Figure 5 brainsci-13-00281-f005:**
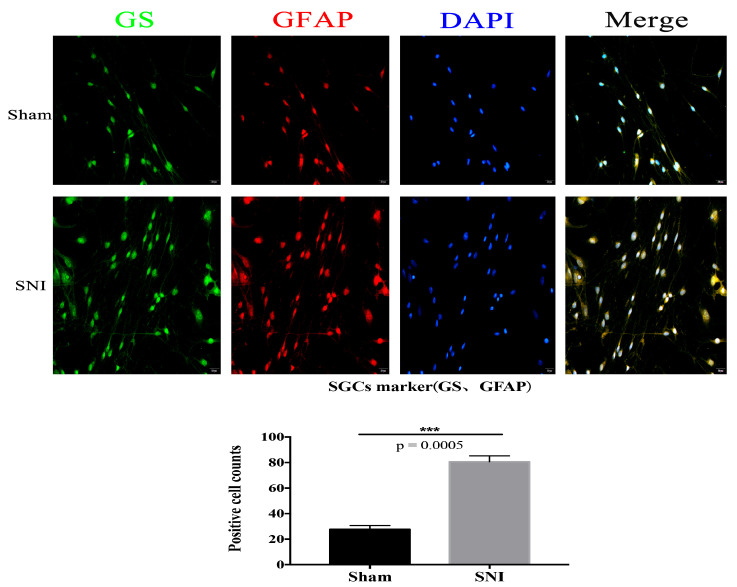
The co-labeling results of GS and GFAP (Scale bar: 20 μm, 40×) (*** *p* = 0.0005).

**Figure 6 brainsci-13-00281-f006:**
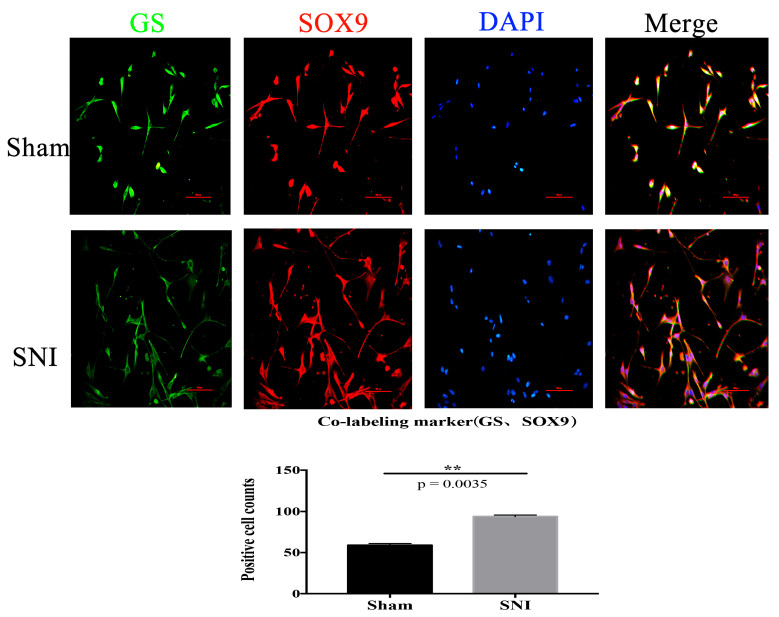
The co-labeling results of GS and Sox9 (Scale bar: 100 μm, 200×) (** *p* = 0.0035).

**Figure 7 brainsci-13-00281-f007:**
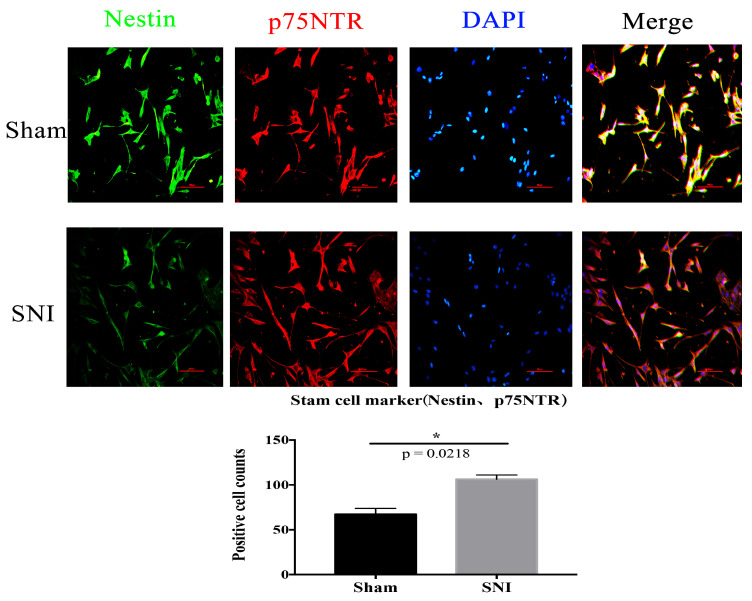
The co-labeling results of Nestin and p75NTR (Scale bar: 100 μm, 200×) (* *p* = 0.0218).

**Figure 8 brainsci-13-00281-f008:**
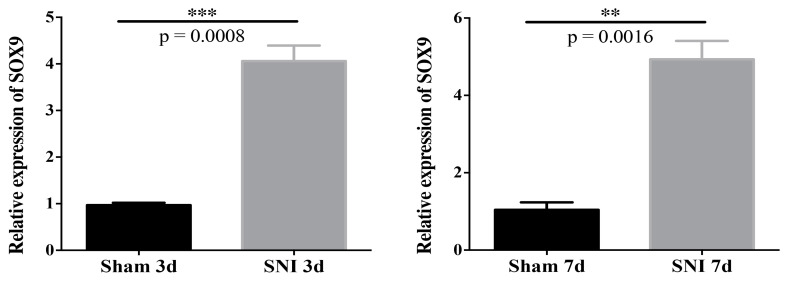
Expression of Sox9 mRNA in SGCs (*** *p* = 0.0008; ** *p* = 0.0016).

**Figure 9 brainsci-13-00281-f009:**
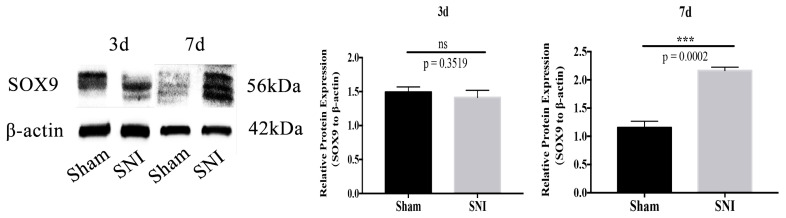
Expression of Sox9 protein in SGCs (ns: no statistical significance; *** *p* = 0.0002).

**Table 1 brainsci-13-00281-t001:** KEGG pathway enrichment analysis.

ID	Description	*p*-Value	Gene ID
mmu00100	Steroid biosynthesis	0.00000000256	Sc5d/Dhcr24/Hsd17b7/Tm7sf2/Lss/Dhcr7/Msmo1/Sqle/Nsdhl/Cyp51
mmu04340	Hedgehog signaling pathway	0.000165308	Kif3a/Btrc/Bcl2/Smurf2/Prkacb/Cdon/Ccnd1/Kif7/Spop/Csnk1d
mmu00900	Terpenoid backbone biosynthesis	0.000339229	Hmgcr/Hmgcs1/Fdps/Acat2/Pcyox1/Ggps1
mmu05014	Amyotrophic lateral sclerosis	0.000622229	Atp5h/Atg14/Chchd10/Atp5b/Atp5j/Atp5c1/Uqcr11/Bcl2/Atp5o/Pik3r4/Wipi2/Optn/Cox6c/Cox7a1/Tpr/Nup107/Nxt2/Cyc1/Atxn2l/Tuba1c/Tnfrsf1a/Xbp1/Seh1l/Tubb2b/Tubb6/Hspa5/Map1lc3a/Dctn2/Tuba1a/Bax
mmu05218	Melanoma	0.000991453	Cdh1/Igf1/Hgf/E2f1/Fgf1/Ccnd1/Gadd45a/Cdkn1a/Bax/Fgf3
mmu04310	Wnt signaling pathway	0.001001666	Ctnnd2/Frzb/Daam2/Map3k7/Btrc/Dvl2/Fzd3/Fzd1/AF366264/Prkacb/Ccnd1/Vangl1/Cxxc4/Siah1b/Tle6/Rnf43/Fzd6
mmu05016	Huntington disease	0.001358182	Atp5h/Ppargc1a/Atg14/Atp5b/Atp5j/Atp5c1/Uqcr11/Slc1a3/Atp5o/Pik3r4/Kcnj10/Wipi2/Sp1/Cox6c/Cox7a1/Cyc1/Tuba1c/Vdac3/Tubb2b/Tubb6/Itpr1/Dctn2/Tuba1a/Cltb/Bax
mmu05132	Salmonella infection	0.001392876	Dynlt3/Map3k7/Vps33a/Bcl2/Wasf3/Rab7b/Dynll2/Tuba1c/Tnfrsf1a/Rhog/Myl12b/Tubb2b/Tubb6/Pik3c2b/Podxl/Dctn2/Casp4/Tuba1a/Nfkbia/Rps3/Casp7/Bax
mmu05226	Gastric cancer	0.00207011	Cdh1/Hgf/E2f1/Dvl2/Fgf1/Fzd3/Bcl2/Fzd1/Ccnd1/Gadd45a/Abcb1a/Cdkn1a/Fzd6/Bax/Fgf3
mmu05012	Parkinson disease	0.002389542	Atp5h/Atp5b/Keap1/Atp5j/Atp5c1/Uqcr11/Atp5o/Prkacb/Cox6c/Cox7a1/Mfn1/Cyc1/Tuba1c/Vdac3/Maoa/Xbp1/Tubb2b/Tubb6/Hspa5/Itpr1/Tuba1a/Bax

**Table 2 brainsci-13-00281-t002:** Top 10 core genes in different algorithms.

Different Algorithms				
	Bottleneck	EPC	Closeness	Readiality
Top 10 core genes	Ank3 ↑	Smurf2 ↓	Ccl2 ↑	Ccl2 ↑
	Cenpf ↑	Ace ↑	Ctgf ↑	Cenpf ↑
	Atf3 ↑	Ccl2 ↑	Hgf ↓	Ctgf ↑
	Gngt2 ↑	Ctgf ↑	Atf3 ↑	Atf3 ↑
	Rab11fip5 ↑	Hgf ↓	Plau ↑	Plau ↑
	Kif22 ↑	Atf3 ↑	Cdkn1a ↑	Cdkn1a ↑
	Kif3a ↓	Cdkn1a ↑	Mmp2 ↑	Mmp2 ↑
	Mmp2 ↑	Kif22 ↑	Igf1 ↓	Igf1 ↓
	Igf1 ↓	Mmp2 ↑	Gm3839 ↑	Sox9 ↑
	Sox9 ↑	Sox9 ↑	Sox9 ↑	Cks2 ↑

↑: Upregulated gene expression; ↓: Downregulated gene expression.

## Data Availability

All the original data in this study can be obtained from the corresponding author upon reasonable request. The GEO dataset GSE120284 contains the original data used in this study, which is freely available.

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
