# Peer review of "Integrative Analysis Reveals the Expression Pattern of SOX9 in Satellite Glial Cells after Sciatic Nerve Injury"

_brainsci, 2023, doi:10.3390/brainsci13020281_

Round 1

Reviewer 1 Report

Dear Author,

Thanks for submitting your research manuscript entitled "Integration analysis of satellite glial cell-related core genes and expression patterns of related transcription factors in mice after sciatic nerve injury".

Before giving my final comments as well as the final revision of this manuscript, the author needs to address the following comments scientifically.
Major concerns:

Please find out the following comments:

·         Manuscript is not in MDPI journal format.

·         The rationale and purpose behind selecting the satellite glial cell-related core genes in sciatic nerve injury is explained very poorly, irrelevant and in incomplete manner throughout the manuscript.

·         Lack of update references with incomplete experimental design is another major concern.  

·         Remove all outdated reference e.g. 2003, 1991, as well as check the reference style as per the MDPI journal format. Check carefully throughout the manuscript.

·         Title and abstract is misleading the reader. Title needs to reframe in simply manner accordingly.

·         Rationale, Selection and evaluation of several cellular and molecular targets in in sciatic nerve injury is very poorly explained, and justified in abstract, intro as well in discussion part.

·         The reviewer found irrational and non-scientific justification in the abstract—introduction and discussion part.

·         Abstract is very poorly written and very confusing. Irrational and fused with repetitions. The reviewer found irrational and non-scientific justification in the abstract—introduction and discussion part.

Example 1: Extremely poor and worst explanation, especially in abstract

A bioinformatics integration analysis was conducted to explore gene expression differences in satellite glial cells in mice after sciatic nerve injury, identify related core genes, and predict their function. The data set of the GSE120284 series was downloaded from the Gene Expression Omnibus profile database, WebGestalt was used to analyze Gene Ontology function and Kyoto Encyclopedia of Genes and Genomes pathway enrichment, Search Tool for the Retrieval of Interacting Genes/Proteins (STRING) analysis and Cytoscape software were used to construct a protein-protein interaction (PPI) network.????????????

Example 2: “In total, 991 differentially expressed genes were identified, including 383 up-regulated and 508 down-regulated genes. Differential genes in biological processes were mainly concentrated in biological regulation, metabolic process, response to stimulus, etc. Differential genes in cell components and molecular function are mostly enriched in membrane, nucleus, protein-containing complex, protein binding, ion binding, nucleic acid binding, nucleotide binding, etc.KEGG analysis showed that the differential genes were mostly enriched in cancer signaling pathways.?????????

Example 3: Conclusion: “In addition, the expression pattern of SOX9 in dorsal root ganglion tissues before and after sciatic nerve injury is similar to that in spinal cord tissues, which makes us curious about the role of SOX9 in the peripheral nervous system.???? What authors want to say? The incomplete justification and scientific correlation is another concern.

-          Remove these types of vague sentences throughout the manuscript: be focused, and just present that team observed.

Highlights 1.For the first time, the key genes in satellite glial cells after sciatic nerve injury were identified by integrated analysis. 2. For the first time, the expression pattern of SOX9 gene in the peripheral nervous system, including dorsal root ganglion tissues and satellite glial cells, was explored.It is suggested that satellite glial cells may have properties similar to stem cells. 3.The expression pattern of SOX9 was similar in central nervous system and peripheral nervous system.

·         The results and discussion are very poorly explained.

Reviewer surprise to see the justification at the end of discussion part “Through in-depth research on the relationship between biological function and protein interaction network and other aspects, the hub gene with greater correlation was screened out. Combined with the relevant background analysis of the project, the expression of hub gene was detected through experiments, providing insights for the study of peripheral glial cells in nerve regeneration”

Author need to directly strike in scientific and readily manner. And simplify whole manuscript directly focus on incidence of actual concern and remove all lines, paragraphs that are saying irrelevant correction etc……..

·         The reviewer feels the author needs to elaborate and justify it with proper citations and strong evidence. The author fails to explain the relevant justification in the introduction as mentioned in the discussion part.

·         A major drawback is a lack of supporting pre-clinical and clinical evidence regarding targeting drugs in sciatic nerve injury.

·         Throughout the manuscript, the main focus is not clear. Complete mismatch of abstract, introduction, results and discussion in concern with effective in nerve injury. Author didn’t justify specific.

Title:

·         Mismatch of title with relevant introduction and conclusive remarks in the conclusion part.

Abstract:

-     The rationale behind this research is not well explained, and several major concerns still constrain the reviewer's enthusiasm for publishing this manuscript.
Introduction:

- The basic literature is not well written and does not even include any literature on alternative approaches with updated references regarding involvement of current drug treatment/techniques used in pathogenesis and development of glial cells in nerve regeneration and related therapies or preventive measurements.

- Authors fail to justify the correlation, and almost irrational and common information is present in the introduction part.

Material and methods:

-     Major drawback is the lack of supporting references and incomplete experimental and paradigms.

- All biochemical parameters are very poorly explained.

- Provide all biochemicals kits numbers along with their city, country in all individual parameters in all expressions, blots, etc.

- In order to support the assessment of all mentioned parameters in his study, the author should provide all the source documents and data he/she has followed for all assays and estimates.

- How was the dosing determined? Dose-responses should be performed.

- How was the sample size determined? Ideally, a priori sample size calculation should be performed to determine the appropriate sample size.
- Normality and variance homogeneity should be assessed across all groups of the same outcome variable and not individual experimental groups. If the data were not normally distributed or variance homogeneity was not met, nonparametric tests need to be performed.
Parametric data should be reported as mean +/- SD, while nonparametric data should be given/displayed as median and interquartile range. Longitudinal data should be analyzed using repeated measures tests.

Results:

-          All results are very poorly explained. Revised all.

-          All blot analysis, and all bar graphs unable to reach. It seems they are copy paste. Provide clear and clean blots, Immunohisto and bar graphs for further revaluation due to their blurriness and there is no clarity for easy understanding. Not acceptable in current form.

-          Re-check stat of figures and confirm either statistical symbol is properly mentioned in graphs or not?

-          Add the scale as well as symbols in immunofluoroscent figures. Here, scale and symbols are not well explained

-          During stat p<0.01 is another major concern. Need to verify. Therefore, provide the supplementary data of all graphs for further verification. Without this, article can’t proceed further.

-          Results need more clarification and significant justification. Differentiating between the outcome and the discussion sections is quite difficult.

-          Convert table 1 (Expression of CD133 in SGC), also in bar graph.

-          High note: Must provide all results description and Use proper statistical reporting: i.e. for the results of each statistical test, the authors should report the statistical test that was applied, the test statistic (e.g. t, U, F, r), degrees of freedom as subscripts to the test statistic, and the exact probability value, including those for normality and variance homogeneity tests. Statistics should be reported in APA format, i.e.: t(df) = value, p = value; F(df1,df2) = value, p = value; r(df) = value, p = value; [chi]2 (df, N = value) = value, p = value; Z = value, p = value.  Include statements on the tests for normality and variance heterogeneity and respective results. If the data were not normally distributed or variance heterogeneity was not met, nonparametric tests need to be applied.

Discussion:

-     To address the outcome of in-vivo measures/results separately and how they correlate with the existing literature, it would be better if the author restructured to take a more critical approach for effective in spinal cord injuries.
-     In the discussion and the conclusion, the aims, rationale, and future perspectives are not evident clearly in relation with in-vitro and in-vivo experimentation.
-     The discussion is usually unorganized at the beginning to address all the observations and evaluate them at the end. It makes the results easier to contextualize and simpler to comprehend.

- Furthermore, a minimal critical analysis should be provided, along with current study limitations as well the future perspective as separate paragraph.

Conclusion:

-          Need to revise the conclusion in a scientific manner. Not accepted in its current form.

-          This reviewer considers that this paper cannot be published in the present form. A detailed revision shortening, ordering and following the commented ideas could improve this interesting paper in a significant manner.

-          Several typewriting mistakes are present and needing correction. This reviewer remains at entire disposal for the next version.

Author Response

Dear editors and reviewers:

Thank you for your professional and rigorous comments and suggestions. I agree with you and accept these suggestions. These suggestions are very valuable and can help me modify my article, enhancing its quality and being highly important for my future research. I have carefully read your suggestions and found that the article needed to be polished at every point. I will upload the revised version and respond to every reviewer’s comment in the cover letter.

Reviewer 1:

  1. Comment: Manuscript is not in MDPI journal format

Response: I have revised the new manuscript according to the format of the journal, and I will re-upload the revised full text to the submission system.

  1. Comment: The rationale and purpose behind selecting the satellite glial cell-related core genes in sciatic nerve injury is explained very poorly, irrelevant and in incomplete manner throughout the manuscript.

Response: In the revised manuscript, we have covered the following: (1) how to screen for Sox9 in DEGs, (2) reasons why Sox9 is important, (3) verifying Sox9 expression in SGCs, and (4) briefly discussing the relationship between Sox9 and SGCs.

  1. Comment:Lack of update references with incomplete experimental design is another major concern. Remove all outdated reference e.g. 2003, 1991, as well as check the reference style as per the MDPI journal format. Check carefully throughout the manuscript.

Response: I have provided the experimental design in the new manuscript. I have deleted and replaced all the outdated references. Furthermore, I have modified the references according to the journal’s format.

  1. Comment:Title and abstract is misleading the reader. Title needs to reframe in simply manner accordingly.

Response: I have modified the title to “Integrative analysis revealed the expression pattern of Sox9 in satellite glial cells after sciatic nerve injury,” and rephrased the abstract.

  1. Comment: Rationale, Selection and evaluation of several cellular and molecular targets in in sciatic nerve injury is very poorly explained, and justified in abstract, intro as well in discussion part.

Response: The first sentence in the “Discussion” section indicates that sciatic nerve injury was selected since it is the most prevalent peripheral nerve injury. “SNI is the most prevalent PNS, characterized by motor and sensory fiber damage[19]". Schwann cells and satellite glial cells are found in the peripheral nervous system. Schwann cells have been reported more frequently following sciatic nerve injury, although satellite glial cells are rare. The final paragraph of the “Discussion" section of the new manuscript summarizes the sciatic nerve injury treatment.

  1. Comment: Abstract is very poorly written and very confusing. Irrational and fused with repetitions. The reviewer found irrational and non-scientific justification in the abstract—introduction and discussion part. Remove these types of vague sentences throughout the manuscript: be focused, and just present that team observed.

Response: I have revised the manuscript to make the “Abstract,” “Introduction,” and “Discission” sections clear to present our findings. Additionally, I have removed vague sentences throughout the new manuscript.

  1. Comment: Remove these types of vague sentences throughout the manuscript: be focused, and just present that team observed. author need to directly strike in scientific and readily manner. And simplify whole manuscript directly focus on incidence of actual concern and remove all lines, paragraphs that are saying irrelevant correction etc……..

Response: I have revised the new manuscript to remove vague sentences and present our findings.

  1. Comment:The reviewer feels the author needs to elaborate and justify it with proper citations and strong evidence. The author fails to explain the relevant justification in the introduction as mentioned in the discussion part. A major drawback is a lack of supporting pre-clinical and clinical evidence regarding targeting drugs in sciatic nerve injury. A major drawback is a lack of supporting pre-clinical and clinical evidence regarding targeting drugs in sciatic nerve injury. Throughout the manuscript, the main focus is not clear. Complete mismatch of abstract, introduction, results and discussion in concern with effective in nerve injury. Author didn’t justify specific.

Response: I have revised the new manuscript to include pre-clinical and clinical evidence regarding targeting drugs in sciatic nerve injury. Its treatment is also summarized in the last part of the “Discussion” section.

  1. Comment:Title Mismatch of title with relevant introduction and conclusive remarks in the conclusion part. The basic literature is not well written and does not even include any literature on alternative approaches with updated references regarding involvement of current drug treatment/techniques used in pathogenesis and development of glial cells in nerve regeneration and related therapies or preventive measurements.

Response: I have modified the title to “Integrative analysis revealed the expression pattern of Sox9 in satellite glial cells after sciatic nerve injury.” I have also included details of the alternate approaches, thus improving the “Introduction” section on glial cells.

  1. Comment:Provide all biochemicals kits numbers along with their city, country in all individual parameters in all expressions, blots, etc.

Response: I have provided the above details in the new manuscript.

  1. Comment:In order to support the assessment of all mentioned parameters in his study, the author should provide all the source documents and data he/she has followed for all assays and estimates.

Response: Source documents and data availability statements have been provided at the end of the new manuscript.

  1. Comment:How was the dosing determined? Dose-responses should be performed.

Response: The dosage of anesthesia for sciatic nerve injury was determined following the references, which have been added “[17Yang,H.,  Zeng,L.,  Liu,Y., Long,Z., Li,Y., Wu,Y. Ultramicro changes of pathology and motor function of sciatic nerve after transverse injury in rats. Chinese Journal of Clinical Rehabilitation. 2003;7:2396-2397. doi: 10.3321/j.issn:1673-8225.2003.17.004 Chinese article”

  1. Comment: How was the sample size determined? Ideally, a priori sample size calculation should be performed to determine the appropriate sample size.

Response: According to the GSE120284 dataset, the experimental groups were: the Sham operation SGCs 3d group and sciatic nerve injury SGCs 3d group, corresponding to the analysis of the dataset. The study added the Sham operation SGCs 7d group and sciatic nerve injury SGCs 7d group for additional verification. Nine replicates were set in each group because the DRG tissues of the opposite L4 were isolated and cultured in the injured group. This was done to ensure the cell volume could carry out subsequent experiments.

  1. Comment: Normality and variance homogeneity should be assessed across all groups of the same outcome variable and not individual experimental groups. If the data were not normally distributed or variance homogeneity was not met, nonparametric tests need to be performed. Parametric data should be reported as mean +/- SD, while nonparametric data should be given/displayed as median and interquartile range. Longitudinal data should be analyzed using repeated measures tests.

Response: The statistical analysis in the “Materials and Methods” section was corrected:Prism software (Ver 7.0, GraphPad Software, San Diego, CA, USA) software was used for data analysis. All data are expressed as mean ± standard deviation (S.D.). Analysis of variance (ANOVA) was used, followed by Bonferroni post-hoc test between groups. P < 0.05 was considered statistically significant.

15.Comment: All results are very poorly explained. Revised all. Re-check stat of figures and confirm either statistical symbol is properly mentioned in graphs or not?

Response: The statistical symbols have been corrected in the revised manuscript.

  1. Comment: Add the scale as well as symbols in immunofluoroscent figures. Here, scale and symbols are not well explained

Response: Scales and symbols in the immunofluorescence figures have been added in the new manuscript.

  1. Comment: During stat p<0.01 is another major concern. Need to verify. Therefore, provide the supplementary data of all graphs for further verification. Without this, article can’t proceed further.

Response: I have corrected the statistics. I'd like to show the raw data.

  1. Comment: Convert table 1 (Expression of CD133 in SGC), also in bar graph.

Response: The other reviewers pointed out that the data of CD133 in this paper is not meaningful and can be deleted. Therefore, the new manuscript will not have the data of CD133.

  1. Comment: High note: Must provide all results description and Use proper statistical reporting: i.e. for the results of each statistical test, the authors should report the statistical test that was applied, the test statistic (e.g. t, U, F, r), degrees of freedom as subscripts to the test statistic, and the exact probability value, including those for normality and variance homogeneity tests. Statistics should be reported in APA format, i.e.: t(df) = value, p = value; F(df1,df2) = value, p = value; r(df) = value, p = value; [chi]2 (df, N = value) = value, p = value; Z = value, p = value. Include statements on the tests for normality and variance heterogeneity and respective results. If the data were not normally distributed or variance heterogeneity was not met, nonparametric tests need to be applied.

Response: I have provided the p values for all statistical tests in the new manuscript.

  1. Comment: To address the outcome of in-vivo measures/results separately and how they correlate with the existing literature, it would be better if the author restructured to take a more critical approach for effective in spinal cord injuries.

Response: Transcription factors may be a new clue for treating the injury. An integrated analysis similar to this study is added in the “Discussion” section of the new manuscript to explore inflammatory factors rather than transcription factors. Still, the specific mechanism of transcription factors needs further research.

  1. Comment: In the discussion and the conclusion, the aims, rationale, and future perspectives are not evident clearly in relation with in-vitro and in-vivo experimentation. Furthermore, a minimal critical analysis should be provided, along with current study limitations as well the future perspective as separate paragraph.

Response: The new manuscript provides the prospects in the last paragraph of the “Discussion” section.

  1. Comment: The discussion is usually unorganized at the beginning to address all the observations and evaluate them at the end. It makes the results easier to contextualize and simpler to comprehend.

Response: The framework for the “Discussion” section has been reworked in the new manuscript.

  1. Comment: Conclusion:Need to revise the conclusion in a scientific manner. Not accepted in its current form.

Response: I have revised the “Conclusion” section in the revised manuscript.

  1. Comment: This reviewer considers that this paper cannot be published in the present form. A detailed revision shortening, ordering and following the commented ideas could improve this interesting paper in a significant manner.

Response: The new manuscript has been revised into a concise version.

  1. Comment: Several typewriting mistakes are present and needing correction. This reviewer remains at entire disposal for the next version.

Response: We apologize for the poor grammar and illogical reasoning in the previous version of the manuscript. A native English speaker has corrected the new manuscript, and we thank you for your comments and suggestions.

Reviewer 2 Report

This study is very informative and well-expressed. I could not find any inadequate information relevant to this study. I would like to congratulate authors.

Minor comment:

-The Figure 1 is not clear. In my opinion, they should have seperated a few of them. There are also same problems with Figures 3 and 4.

Author Response

Dear editors and reviewers:

Thank you for your professional and rigorous comments and suggestions. I agree with you and accept these suggestions. These suggestions are valuable and can help me modify article, enhancing its quality and being highly important for my future research. I have carefully read your suggestions and found that the article needed to be polished at every point. I will upload the revised version and respond to every reviewer’s comment in the cover letter.

Reviewer 2:

Comment:The Figure 1 is not clear. In my opinion, they should have seperated a few of them. There are also same problems with Figures 3 and 4.

Response: To improve clarity, the new manuscript has been reformatted. We omitted flow cytometry CD133 from the results after examining the reasoning of the paper, including the introduction and discussion sections, because the results of this part were redundant and lengthy in the paper, and the manuscript needed to be made simpler. A native English speaker has modified the English language of this paper.

Reviewer 3 Report

This manuscript presents results of original investigation in the field of bioinformatic analysis of gene expression data related to peripheral nerves. The topic is very interesting and current. Methods applied in the research are appropriate; results are clearly presented and critically discussed. The cited references are adequate and contemporary. In order to explore gene expression differences in satellite glial cells in mice after sciatic nerve injury, identify related core genes, and predict their function, the authors performed bioinformatic integration analysis. They analyzed the data set of the GSE120284 series available online using several software tools with the aim to elucidate gene ontology function and construct protein – protein interaction network. Total number of 991 differentially expressed genes were identified (383 up – regulated and 508 down – regulated); majority of these genes have a role in biological regulation, metabolic process, response to stimulus. According to KEGG (Kyoto Encyclopedia of Genes and Genomes) pathway enrichment analysis, differential genes were mostly enriched in cancer signaling pathways. In addition, using different algorithms eight hub genes were identified, including Sox9 gene. Protein analyses confirmed that expression of SOX9 in dorsal root ganglion and satellite glial cells was more obvious than before injury. Interesting enough, the mentioned above expression pattern of SOX9 was similar to that in spinal cord tissues, which raises the question about the role and mechanism of action of SOX9 in the peripheral nervous system. In this manuscript for the first time the key genes in satellite glial cells after sciatic nerve injury were identified by integrated analysis. Also, for the first time was explored the expression pattern of SOX9 gene in the peripheral nervous system, including dorsal root ganglion tissues and satellite glial cells; the results indicated that satellite glial cells may have properties similar to stem cells. Having in mind all above this manuscript has importance for both basic science and medical practice.

Author Response

Dear editors and reviewers:

Thank you for your professional and rigorous comments and suggestions. I agree with you and accept these suggestions. These suggestions are valuable and can help me modify article, enhancing its quality and being highly important for my future research. I have carefully read your suggestions and found that the article needed to be polished at every point. I will upload the revised version and respond to every reviewer’s comment in the cover letter.

Reviewer 3

 I appreciate the reviewers’ acknowledgment of this study. I have revised the manuscript to make it publication-ready. (1) The core genes related to the background of the paper have been screened, verified, and discussed. (2) Simplify the manuscript and incorporate relevant, high-quality literature to use the article convincingly. (3) Improve picture layout and quality. (4) Improve language quality and structure. Kindly review these changes in the new manuscript.

Reviewer 4 Report

1. The summary is very technical, and should be more general, describing the paper goal and main results. 

2. The Introduction require more references, for example to claims such “satellite glial cells are similar to astrocytes in the CNS, or “A variety of stressors can trigger satellite glial cell activation. 

3. The statement “The PNS does not possess a classical stem cell population” is not correct as the Neural Crest cells give rise to the PNS population.  

4. In one of the papers cited, Weider M et al, the statement is not correct- it does not describe reprogramming of satellite glial cells into oligodendrocytes. 

5. Figure 2- There are no quantifications or statistics of the images. Bar graphs are missing.  

6. The results part does not include any hypothesis or explanation of the experiments rational. 

7. There are very inaccurate statements, such as: “SGCs are similar to adult rat SCs with regard to transcription and morphology, forming a sheath around the neuronal body”. SC are surrounding the axons and not the neuronal cell body. 

8. Since in this paper they analyze a published dataset, they should better mention what was discovered in the original paper (Jager, 2020) and what is their analysis addition. 

9. Figures are in low resolution and can’t read the text. Images are pixelated. 

The rational for the experiments is not explained- in fig.2 - why cells were stained for those markers?, fig.3- why CD34, CD133 and CD45 were tested?  

10. Why was Sox9 chosen as a marker in SGC? The authors do not present any evidence that Sox9 is expressed in SGC. A co-staining with a SGC marker is required.  

11. The paper is poorly written with lots of grammar mistakes.

Author Response

Dear editors and reviewers

Thank you for your professional and rigorous comments and suggestions. I agree with you and accept these suggestions. These suggestions are valuable and can help me modify my article, enhancing its quality and being highly important for future research. I have carefully read your suggestions and found that the article needed to be polished at every point. I will upload the revised version and respond to every reviewer’s comment in the cover letter.

Reviewer 4:

Comment:1. The summary is very technical, and should be more general, describing the paper goal and main results. 

Response: In the new manuscript, I have revised the abstract according to the journal’s format.

Comment:2. The Introduction require more references, for example to claims such “satellite glial cells are similar to astrocytes in the CNS, or “A variety of stressors can trigger satellite glial cell activation”. 

Response: I have revised the “Introduction” section of the new manuscript and updated the references.

Comment:3. The statement “The PNS does not possess a classical stem cell population” is not correct as the Neural Crest cells give rise to the PNS population.  

Response: I have read the literature again, and I have corrected the inaccurate statements according to your suggestion. These inaccurate sentences have been deleted in the new manuscript.

Comment:4. In one of the papers cited, Weider M et al, the statement is not correct- it does not describe reprogramming of satellite glial cells into oligodendrocytes. 

Response: Thank you for pointing this out. I have corrected this sentence in the new manuscript.

Comment:5. Figure 2- There are no quantifications or statistics of the images. Bar graphs are missing.  

Response: I have added scales to the immunofluorescence figures in the new manuscript.

Comment:6. The results part does not include any hypothesis or explanation of the experiments rational. 

Response: We have added an explanation of the rationale of the experiment in the “Results” section of the new manuscript.

Comment:7. There are very inaccurate statements, such as: “SGCs are similar to adult rat SCs with regard to transcription and morphology, forming a sheath around the neuronal body”. SC are surrounding the axons and not the neuronal cell body. 

Response: I have corrected the above inaccurate statements in the new manuscript.

Comment:8. Since in this paper they analyze a published dataset, they should better mention what was discovered in the original paper (Jager, 2020) and what is their analysis addition. 

Response: I have added a brief analysis of similar studies, including “Jager, 2020” to the last paragraph of the “Discussion” section of the new manuscript. These studies have explored the inflammatory factors rather than the transcription factors.

Comment:9. Figures are in low resolution and can’t read the text. Images are pixelated. 

The rational for the experiments is not explained- in fig.2 - why cells were stained for those markers?, fig.3- why CD34, CD133 and CD45 were tested?  

Response: The new manuscript has been reformatted along with the figures. After revising the article, we thought the CD133 data was not relevant to the manuscript; therefore, we deleted it in the new manuscript.

Comment:10. Why was Sox9 chosen as a marker in SGC? The authors do not present any evidence that Sox9 is expressed in SGC. A co-staining with a SGC marker is required.  

Response: In the new manuscript, we specified that SGC was co-stained with SOX9, and GS was used as the SGC marker.

Comment:11. The paper is poorly written with lots of grammar mistakes.

Response: We apologize for the poor grammar and illogical reasoning in the previous version of the manuscript. A native English speaker has corrected the new manuscript, and we thank you for your comments and suggestions.

Round 2

Reviewer 1 Report

Dear Author, 

After careful revision, the revised manuscript can be proceed further for publication.